# Detection and treatment of mental disorders in patients with coronary heart disease (MenDis-CHD): A cross-sectional study

**Samia Peltzer**[1]*, **Hendrik Müller**[2], **Ursula Köstler**[3], **Frank Schulz-Nieswandt**[3], **Frank Jessen**[2,4], **Christian Albus**[1], on behalf of the CoRe-Net study group[¶]

1 Department of Psychosomatics and Psychotherapy, Faculty of Medicine, University Hospital Cologne, University of Cologne, Cologne, North Rhine-Westphalia, Germany, 2 Department of Psychiatry and Psychotherapy, Faculty of Medicine, University Hospital Cologne, University of Cologne, Cologne, North Rhine-Westphalia, Germany, 3 Faculty of Management, Economics and Social Sciences, University of Cologne, Cologne, North Rhine-Westphalia, Germany, 4 German Center for Neurodegenerative Diseases (DZNE), Bonn, North Rhine-Westphalia, Germany

¶ Membership of the Cologne Research-Network study group (CoRe-Net; collaboration group) is provided in the Acknowledgments.
* samia.peltzer@uk-koeln.de

**Data Availability Statement:** The data underlying the results presented in the study cannot be shared publicly because of ethical restrictions imposed by

## Abstract

Mental disorders (MD) are associated with an increased risk of developing coronary heart disease (CHD) and with higher CHD-related morbidity and mortality. There is a strong recommendation to routinely screen CHD patients for MDs, diagnosis, and treatment by recent guidelines. The current study aimed at mapping CHD patients' (1) state of diagnostics and, if necessary, treatment of MDs, (2) trajectories and detection rate in healthcare, and (3) the influence of MDs and its management on quality of life and patient satisfaction. The design was a cross-sectional study in three settings (two hospitals, two rehabilitation clinics, three cardiology practices). CHD patients were screened for MDs with the Hospital Anxiety and Depression Scale (HADS), and, if screened-positive, examined for MDs with the Structured Clinical Interview for DSM-IV (SCID-I). Quality of Life (EQ-5D), Patient Assessment of Care for Chronic Conditions (PACIC), and previous routine diagnostics and treatment for MDs were examined. Descriptive statistics, Chi-squared tests, and ANOVA were used for analyses. Analyses of the data of 364 patients resulted in 33.8% positive HADS-screenings and 28.0% SCID-I diagnoses. The detection rate of correctly pre-diagnosed MDs was 49.0%. Physicians actively approached approximately thirty percent of patients on MDs; however, only 6.6% of patients underwent psychotherapy and 4.1% medication therapy through psychotherapists/psychiatrists. MD patients scored significantly lower on EQ-5D and the PACIC. The state of diagnostic and treatment of comorbid MDs in patients with CHD is insufficient. Patients showed a positive attitude towards addressing MDs and were satisfied with medical treatment, but less with MD-related advice. Physicians in secondary care need more training inadequately addressing mental comorbidity.

the Ethics Commission of Cologne University's Faculty of Medicine. Data access queries may be directed to Dr. Guido Grass, the Head of the Office of the Ethics Committee of the Medical Faculty of the University of Cologne (contact via tel.: +49 221 478 87916 or via email: ek-med@uni-koeln.de), or to Dr. Holger Pfaff at the CoRe-Net data trust center (contact via tel.: +49 221 478 97100 or via email: holger.pfaff@uk-koeln.de).

**Funding:** All authors disclosed receipt of the following financial support for the research, authorship, and/or publication of this article: This work was supported by the BMBF (Federal Ministry of Research and Education [grant number: 01GY1606]. URL: https://www.bmbf.de/ The funders had no role in study design, data collection and analysis, decision to publish, or preparation of the manuscript.

**Competing interests:** The authors have declared that no competing interests exist.

## Introduction

According to the Global Burden of Disease Study, cardiovascular disease, especially coronary heart disease (CHD), will be a leading burden of disease in the coming decades and remains the leading cause of mortality in Europe [1]. Mental disorders (MD), such as major depression and anxiety disorders, are also a significant contributor to the global burden of disease. 14.3% of all deaths worldwide are attributable to MDs [2]. MDs have an enormous impact on healthcare costs, are associated with an increased risk of developing CHD, and worsen prognosis in established disease [3]. Depression is associated with a nearly twofold risk for developing CHD (*OR* 1.6–1.9) and with higher CHD-related morbidity and mortality (*OR* 1.6–2.4) [4]. After a myocardial infarction, almost 30% of patients experience depressive symptoms, 20% fulfilled the criteria for depressive disorders. During the next two years after a cardiac event, CHD patients with a comorbid depression disorder have two-fold greater mortality. The risk of dying can increase six-fold when depressive symptoms are severe [5]. Prevalence of anxiety is associated with a significant risk of cardiovascular mortality and CHD (*OR* 1.41) [6], especially phobic anxiety and panic disorders [5]. Approximately 39% of women and 22% of men experienced anxious symptoms 1.4 years after hospitalization for CHD [7].

The negative impact of MDs on the incidence and prognosis of CHD is mediated by behavioral and psychobiological mechanisms. Firstly, MDs often acts as a barrier to treatment adherence and reluctance to lifestyle change. Secondly, mechanisms like autonomic nervous system dysfunction, dysregulation of the hypothalamic-pituitary-adrenal axis, and proinflammatory and prothrombotic states directly interfere with the etiopathogenesis of CHD [8, 9]. Furthermore, it is assumed that these patients show a reduced health-related quality of life [3] and satisfaction [10].

Consequently, there is a strong recommendation of routine screening for MDs, adequate diagnosis, and treatment in person at risk for CHD and with clinically manifest CHD by recent national and international guidelines [3, 7, 11]. However, only a few studies have been conducted on the topic of the quality of healthcare concerning the screening and treatment of CHD patients with comorbid MDs. Adherence to CHD guidelines in physicians seems to be generally low, even when guidelines predominantly comprise essential somatic recommendations [12]. From a clinical perspective, it is unlikely that physicians regularly screen for MDs due to restraints in daily routine, e.g., lack of time, competence, or reimbursement [13, 14]. A study that examined various care management processes for depression and other chronic diseases found that physicians in primary care used significantly fewer procedures to detect and treat depression than for somatic diseases and that no adequate depression management was provided [15]. Feinstein et al. [14] showed that half of the examined physicians were unaware that depression itself could be a severe risk factor to CHD, 79% used no screening method to diagnose depression, and 71% asked less than half their patients about depression. Thombs et al. [16] found that screening for depression seems to be beneficial only if it results in a correct diagnosis and if appropriate treatment is available. Otherwise, uncritical use of psychometric screening tools could lead to over-diagnosing MDs, and consequently, inappropriate treatment and high costs for the healthcare system. These issues can be assessed with the measurement of patients' health trajectories. Health trajectories provide information about a patient's health status at any specified time, with the possibility of an included endpoint of interest [17]. In this study, we define the term 'trajectories' as the patient's previous healthcare use and days of sick leave within the last twelve months. Hence, for the quality of care, it is essential not only to look at screening per se but also on whether MDs are addressed within the physician-patient interaction, diagnostic and treatment procedures [16], as well as in patients' trajectories within the healthcare system. Research on the quality of care and

trajectories within the healthcare system of CHD patients with MDs has only recently begun to emerge. Depressive symptoms have been linked to the prediction of mortality within twelve months. At the same time, depressive symptoms persisted in clinically (85%) and subclinically (47%) depressed patients for at least one year [18], indicating that depressive symptoms after a CHD-related incident bear the potential to limit patients' quality of life in the long term. This conclusion is supported by the findings of Palacios et al. [19], who looked at the development over time of five groups, differing in severity of MD-related symptoms. Patients from the 'chronic high' and 'worsening' group had significantly higher healthcare costs over patients who were characterized by less symptom severity over the course of time [19]. Up until now, most studies have limited themselves to assessing anxiety and depression in CHD patients. The current study investigates a broader diagnostic spectrum of MDs in CHD patients with the SCID-I (e.g., PTSD, addiction disorder, bipolar disorder, etc.). Furthermore, MenDis-CHD aimed to explore (1) the current state of diagnostics and treatment of MD and non-MD in CHD patients, (2) patients' trajectories and detection rate in secondary care, and (3) implications of mental comorbidity and its management on quality of life and patient satisfaction. Following the literature [4–7], it was expected that approximately 30% of CHD patients would experience depressive symptoms, and 20% would receive a clinical MD diagnosis on SCID-I. When it comes to patients' trajectories, it was expected that MD patients would more often utilize the healthcare system by, e.g., being hospitalized more frequently or being ill more regularly than non-MD patients. Furthermore, it was hypothesized that a significant amount of MD cases would not be detected by physicians in secondary care [14]. At last, it was expected that patients with MDs would experience significantly lower quality of life and satisfaction on psychological and physiological dimensions than non-MD patients [3, 10].

## Materials and methods

### Theoretical framework and research platform

MenDis-CHD is one of the three current projects of the 'Cologne Care Research and Development Network' (CoRe-Net) [20] in Cologne, Germany, and is funded by the Federal Ministry of Education and Research (BMBF). MenDis-CHD was approved by the Ethics Commission of Cologne University's Faculty of Medicine (committee's reference number: 17–220) on Sept 26th, 2017. This study was conducted according to the principles expressed in the Declaration of Helsinki. Trails registration number: German clinical trials register (Deutsches Register Klinischer Studien, DKRS), i.e., Registration Number: DRKS00012434, date of registration: May 11th, 2017. URL: https://www.drks.de/drks_web/ navigate.do?navigationId = trial. HTML&TRIAL_ID = DRKS00012434. For the study protocol, see [21].

### Participants

Participants were included if they were at least 18 years old, had an angiographically documented CHD, were in treatment (e.g., stable angina pectoris, myocardial infarction), had sufficient German language skills, were patients in one of the cohort settings, and were able to give informed consent. Patients were excluded if they had severe or instable physical or mental conditions.

### Assessments

**Sociodemographic and clinical characteristics.** Sociodemographic and clinical data were assessed by questionnaire, comprising age, gender, marital status, professional status, and medical history. The severity of CHD was assessed from the medical chart and contained i.a., NYHA (New York Heart Association; schema for classification of heart diseases according to

their degree of severity), cardiac events in the previous medical history (e.g., myocardial infarction), cardiac surgeries (e.g., bypass surgery), congestive heart failure, percutaneous coronary intervention (PCI), and left ventricular ejection fraction.

**Current state of psychosomatic support, diagnostics, and treatment.** The patient's state of psychosomatic support, previous diagnostics, and treatments were determined using a self-developed 51 items questionnaire. All recruited patients had to answer to the questionnaire, but patients without MDs could skip several questions. The section 'state of psychosomatic support contained five items such as 'yes or no' questions: 'Do you talk with your physician about psychosocial problems?' and multiple-choice questions such as: 'How often did you talk with your physician about mental problems?' with the answer possibilities: 'always (every time) / often (every second appointment) / sometimes / seldom / only once / never.' Example questions for the other sections were: 'Have you had a psychological/psychiatric examination?' for the section 'diagnostics', and 'Are you currently receiving psychotherapy?' for the section 'treatment'. This questionnaire was developed within this study and has yet to be validated.

The *Hospital Anxiety and Depression Scale* (HADS; [22, 23]) was used as a screening tool for depression and anxiety symptoms on two separate scales. If patients scored eight and above in either the depression or anxiety scale, the screening was considered as 'positive'. Participants with a 'positive' HADS result were further assessed with the SCID-I to detect a suspected MD (*Structured Clinical Interview for DSM-IV;* [24]).

**Patients' trajectories and detection rate.** Patients' trajectories contained in total five questions within the self-developed 51 items questionnaire regarding frequency of consultation of general practitioner (GP), cardiologists and psychotherapists in the last four weeks and in the last twelve months, the frequency of hospital stay (due to heart disease or other diseases) in the last twelve months and days of sick leave in the last twelve months. An example question was: 'How many days have you been unable to work in the last twelve months?'.

Detection rate of MDs in healthcare was assessed with the help of the medical charts of the patients. Patients were classified with currently having MD or not based on their prior diagnoses. During the study, patients then were screened for MDs with the SCID-I. Subsequently, patients were labelled as either (1) incorrectly pre-diagnosed with MDs, (2) incorrectly pre-diagnosed without MDs, (3) correctly pre-diagnosed without MDs, or (4) correctly pre-diagnosed with MDs.

**Quality of life and patient satisfaction.** Quality of life was assessed with the *EURO-Quality of Life 5D-3L* questionnaire (EQ-5D; [25]) on five dimensions (mobility, self-care, usual activities, pain and discomfort, anxiety, and depression). Measurement was conducted with a 3-point Likert scale. Raw scores were transformed in unison with the official EuroQoL guidelines into a *Visual Analogue Scale* (VAS; details see [26]). The higher the EQ-5D score, the lower the health-related quality of life. Also, the EQ-5D contained a visual analog scale with the points 'the best health condition you can imagine' and 'the worst health condition you can imagine', which was used as a quantitative individual measure scale.

The *Patient Assessment of Care for Chronic Conditions* (PACIC; [27]) evaluated patient-healthcare team interactions and aspects of self-management support. The PACIC has five subscales, which measure: patient activation, delivery system design/decision support, goal setting, problem-solving/contextual counseling, and follow-up/coordination. The higher the mean total score, the better the team interactions and support. A modified version of the PACIC was used in this study. The original version of the short-form PACIC uses an 11-point scale, uniformly ranging from 0% to 100%. Our study utilized a 5-point-Likert scale, assessing patients' satisfaction regarding the topics mentioned above (this was advised by one of the developers of the PACIC for having less room for interpretation). Questions concerning the validation of this combination of questions and measurement techniques will be addressed in a separate manuscript.

## Procedures

Patients were recruited in two cardiologic hospital departments, three practices, and two rehabilitation clinics in Cologne, Germany. The recruitment phase took place between Jan 15[th], 2018, and Mar 29[th], 2019. We used the clustered sampling technique, where two hospitals (center one: 'hospitals'), two rehabilitation clinics (center two: 'rehabilitation clinics'), and three cardiology practices (center three: 'practices') were used as sampling units. As in single-stage cluster sampling, all eligible patients of the chosen sampling units were then included in the study if they wanted to participate and fulfilled the inclusion criteria.

Patients were screened for eligibility and documented in a screening log. Patients who fulfilled the inclusion criteria were adequately instructed about the procedure of the study, gave written informed consent, and were handed out the questionnaire-set which they filled in. If the HADS was positive, a second appointment was arranged to perform the SCID-I. All researchers were trained in applying the SCID-I and experienced in conducting it. If patients were diagnosed with MD, they were offered immediate psychological support and further information on psychotherapeutic treatment such as addresses of psychosomatic/ psychological ambulances, or psychotherapists.

## Data analysis

All presented data were analyzed with *IBM SPSS Statistics 22* [28]. The dataset was controlled for outliers, missing values, and implausible values. When cases with either missing or impossible values were identified, we compared these values to the equivalent 'paper and pencil' versions for correction. When outliers were found on a dependent variable, the analysis was conducted twice: once with and once without the outlier to verify that the outlier did not influence the result inadequately. No outlier had to be removed. Example: some patients required pervasive medical care, which resulted in them having more regular contact with physicians than the rest of the sample.

Multiple variables were created from existing variables to reduce the number of value levels for factorial variables. For each analysis, we tested a priori if the respective statistical assumptions were violated: for ANOVA: Independence of errors, homogeneity of variance between groups, and normality of residuals within groups. Weighted means approach based on the size of each factor level was used for unequal cell sizes in variance analyses. To do this, weighted effect codes were used. Minor violations for normality of residuals were found. However, Welch's F (a robust estimate) backed the prior obtained results. Both assumptions for Pearson's Chi-Squared test (i.e., both variables should be measured at an ordinal or nominal level, and both variables should consist of two or more categorical, independent groups) were controlled for and met.

The current state of diagnostics and treatment of MD and non-MD in CHD patients (research aim one) was investigated using Pearson's Chi-squared tests to determine whether there was a statistically significant difference between expected and observed frequencies. See Table 2 for all items used to address this aim. Research aims two (Patients' trajectories and detection rate in secondary care) and three (implications of mental comorbidity and its management on quality of life and patient satisfaction) were examined using Analyses of Covariance (ANCOVAs). (Not) Having MD served as between-subjects factor. The displayed variables in Table 3 were used as dependent variables. In addition to the research aims we also investigated sociodemographic data and clinical characteristics of the cohort (see Table 1). The threshold for significance was set at $\alpha$ = .05. P-values from Tables 2 and 3 were corrected for confounding variables, i.e., age, gender, and NYHA score.

**Table 1. Overview of sociodemographic and clinical characteristics.**

| | Patients with MD | | Patients without MD | | P-value | Total | |
|---|---|---|---|---|---|---|---|
| | N = 102, n (%) | | N = 262, n (%) | | | N = 364, n (%) | |
| **Sociodemographic data** | | | | | | | |
| **Gender** | | | | | .002 | | |
| Male | 60 | 58.8 | 198 | 75.6 | | 258 | 70.9 |
| Female | 42 | 41.2 | 64 | 24.4 | | 106 | 29.1 |
| **Age** | | | | | .001 | | |
| 35–49 years | 9 | 8.8 | 15 | 5.7 | | 24 | 6.6 |
| 50–59 years | 30 | 29.4 | 59 | 22.5 | | 89 | 24.5 |
| 60–69 years | 39 | 38.2 | 72 | 27.5 | | 111 | 30.5 |
| 70–79 years | 10 | 9.8 | 80 | 30.5 | | 90 | 24.7 |
| 80–95 years | 14 | 13.7 | 36 | 13.7 | | 50 | 13.7 |
| Marital status | | | | | .292 | | |
| Living together | 70 | 68.6 | 194 | 74.1 | | 264 | 72.5 |
| Living alone | 32 | 31.4 | 68 | 25.9 | | 100 | 27.5 |
| Professional qualification [a] | | | | | .245 | | |
| None | 14 | 13.7 | 18 | 6.9 | | 32 | 8.8 |
| Apprenticeship | 49 | 48.0 | 127 | 48.5 | | 176 | 48.4 |
| Vocational school | 14 | 13.7 | 44 | 16.8 | | 58 | 15.9 |
| College/university | 15 | 14.7 | 51 | 19.5 | | 66 | 18.1 |
| Other | 10 | 9.8 | 22 | 8.4 | | 32 | 8.8 |
| Retired | 48 | 47.1 | 143 | 54.6 | .197 | 191 | 52.5 |
| **Clinical characteristics** | | | | | | | |
| Somatic comorbidity [a] | | | | | [b] | | |
| Peripheral arterial disease | 9 | 8.8 | 24 | 9.2 | | 33 | 9.1 |
| Congestive heart failure | 25 | 24.5 | 77 | 29.4 | | 102 | 28.0 |
| Transient ischemic attack/stroke | 5 | 4.9 | 22 | 8.4 | | 27 | 7.4 |
| Cancer | 2 | 1.9 | - | - | | 2 | 0.5 |
| Left ventricular ejection fraction | | | | | .597 | | |
| > 40% | 82 | 80.4 | 204 | 77.9 | | 286 | 78.6 |
| ≤ 40% | 20 | 19.6 | 58 | 22.1 | | 78 | 21.4 |
| **NYHA** | | | | | .001 | | |
| NYHA I | 26 | 25.5 | 101 | 38.6 | | 127 | 34.9 |
| NYHA II | 41 | 40.2 | 115 | 43.9 | | 156 | 42.9 |
| NYHA III | 35 | 34.3 | 46 | 17.6 | | 81 | 22.3 |
| Patients with both myocardial infarction and PCI intervention | 62 | 60.8 | 145 | 55.3 | .886 | 207 | 56.9 |
| Only PCI intervention | 85 | 83.3 | 221 | 84.4 | .960 | 306 | 84.1 |
| Bypass surgery | 21 | 20.6 | 47 | 17.9 | .560 | 68 | 18.7 |
| Cardiac valve surgery | 6 | 5.9 | 18 | 6.9 | .733 | 24 | 6.6 |
| **HADS Anxiety** | | | | | < .001 | | |
| Positive (≥ 8) | 68 | 66.7 | 18 | 6.9 | | 86 | 23.6 |
| **HADS Depression** | | | | | < .001 | | |
| Positive (≥ 8) | 55 | 53.9 | 7 | 2.7 | | 62 | 17.0 |
| **HADS Total** | | | | | < .001 | | |
| Positive (sum score ≥ 14) | 102 | 100.0 | 21 | 8.0 | | 123 | 33.8 |

Abbreviations: MD, Mental Disorder; NYHA, New York Heart Association; PCI, Percutaneous Coronary Intervention; HADS, Hospital Anxiety and Depression Scale.

[a]Note: Multiple-choice question.

[b]Note: At least one cell was too small for the appropriate analysis.

**Table 2. Overview of results of MD and non-MD patients regarding the current state of psychosomatic support, diagnostics, and treatment.**

| | Patients with MD | | Patients without MD | | P-value | Total | |
|---|---|---|---|---|---|---|---|
| | N = 102, *n* (%) | | N = 262, *n* (%) | | | N = 364, *n* (%) | |
| **Current state of psychosomatic support** | | | | | | | |
| **Talking with the physician about psychosocial problems** | **62** | **60.8** | **99** | **37.8** | **< .001** | **161** | **44.2** |
| Frequency of talking with the physician about psychosocial problems | | | | | .427 | | |
| Always (on every doctor's appointment) | 13 | 12.8 | 14 | 5.3 | | 27 | 7.4 |
| Often (on every second doctor's appointment) | 11 | 10.8 | 16 | 6.1 | | 27 | 7.4 |
| Sometimes | 25 | 24.5 | 43 | 16.4 | | 68 | 18.7 |
| Seldom | 7 | 6.9 | 22 | 8.4 | | 29 | 8.0 |
| Only once | 4 | 3.9 | 2 | 0.8 | | 6 | 1.6 |
| Never | 1 | 0.9 | 6 | 2.3 | | 7 | 1.9 |
| Actively approached by the physician on MDs | 42 | 42.2 | 72 | 27.5 | .641 | 114 | 31.3 |
| Found it appropriate to have been asked by the physician | 42 | 42.2 | 69 | 26.3 | .594 | 111 | 30.5 |
| **Diagnostics** | | | | | | | |
| **Own perception of decline of mental well-being** | **56** | **54.9** | **26** | **9.9** | **< .001** | **82** | **22.5** |
| **Asked by others about the decline of mental well-being** | **32** | **31.4** | **15** | **5.7** | **.013** | **47** | **12.9** |
| If yes, by whom [a] | | | | | [b] | | |
| General practitioner | 12 | 11.8 | 3 | 1.2 | | 15 | 4.1 |
| Clinic | - | - | 2 | 0.8 | | 2 | 0.5 |
| Rehabilitation clinic | 4 | 3.9 | 3 | 1.2 | | 7 | 1.9 |
| Cardiologist | 2 | 1.9 | 2 | 0.8 | | 4 | 1.1 |
| Family & acquaintances (three categories) | 48 | 47.1 | 26 | 9.9 | | 74 | 20.3 |
| Others | 5 | 4.9 | - | - | | 5 | 1.4 |
| **Psychological/psychiatric examination carried out** | **46** | **45.1** | **6** | **2.3** | **< .001** | **52** | **14.3** |
| **Diagnosed positively MD symptoms** | **46** | **45.1** | **18** | **6.9** | **< .001** | **64** | **17.6** |
| If yes, by whom [a] | | | | | [b] | | |
| General practitioner | 10 | 9.8 | 1 | 0.4 | | 11 | 3.0 |
| Cardiologist | - | - | - | - | | - | - |
| Psychiatrist/psychotherapist | 28 | 27.5 | 2 | 0.8 | | 30 | 8.2 |
| Neurologist | 5 | 4.9 | 1 | 0.4 | | 6 | 1.6 |
| Other | 6 | 5.9 | - | - | | 6 | 1.6 |
| **Referral for additional diagnostics** | **30** | **29.4** | **3** | **1.2** | **< .001** | **33** | **9.1** |
| If yes, to whom [a] | | | | | .021 | | |
| Psychiatrist/psychotherapist | 25 | 24.5 | 2 | 0.8 | | 27 | 7.4 |
| Neurologist | 4 | 3.9 | 3 | 1.2 | | 7 | 1.9 |
| Other | 1 | 0.9 | - | - | | 1 | 0.3 |
| Type of examination [a] | | | | | [b] | | |
| Questionnaires | 16 | 15.7 | 1 | 0.4 | | 17 | 4.7 |
| Tests | 14 | 13.7 | 4 | 1.5 | | 18 | 4.9 |
| Physical examination | 11 | 10.8 | 3 | 1.2 | | 14 | 3.8 |
| CT/MRT | 6 | 5.9 | 4 | 1.5 | | 10 | 2.7 |
| Others | 12 | 11.8 | 2 | 0.8 | | 14 | 3.8 |
| **Patient's knowledge about MD diagnosis** | **41** | **40.2** | **8** | **3.1** | **< .001** | **49** | **13.5** |
| Type of diagnosis [a] | | | | | < .001 | | |
| Depression | 29 | 28.4 | - | - | | 29 | 8.0 |
| Anxiety disorder | 12 | 11.8 | - | - | | 12 | 3.3 |
| Other | 2 | 1.9 | 5 | 1.9 | | 7 | 1.9 |

*(Continued)*

**Table 2.** (Continued)

| | Patients with MD | | Patients without MD | | P-value | Total | |
|---|---|---|---|---|---|---|---|
| | N = 102, n (%) | | N = 262, n (%) | | | N = 364, n (%) | |
| **Receiving an explanation of what the MD diagnosis means for further treatment** | **28** | **27.5** | **4** | **1.5** | **.001** | **32** | **8.8** |
| **Treatment** | | | | | | | |
| Was a treatment for MDs recommended? | 37 | 36.3 | 4 | 1.5 | .176 | 41 | 11.3 |
| If yes, what treatment | | | | | | | |
| Medication | 2 | 1.9 | 1 | 0.4 | | 3 | 0.8 |
| Psychotherapy | 22 | 21.6 | 4 | 1.5 | | 26 | 7.1 |
| Both | 9 | 8.8 | 1 | 0.4 | | 10 | 2.7 |
| Other | 1 | 0.9 | 1 | 0.4 | | 2 | 0.5 |
| **Currently undergoing psychotherapy** | **22** | **21.6** | **2** | **0.8** | **.003** | **24** | **6.6** |
| Perceived improvement through psychotherapy | 18 | 17.7 | 3 | 1.2 | .750 | 21 | 5.8 |
| **Currently undergoing medication therapy** | **13** | **12.8** | **2** | **0.8** | **.020** | **15** | **4.1** |
| Perceived improvement through medication therapy | 12 | 11.8 | 1 | 0.4 | .078 | 13 | 3.6 |

Abbreviations: MD, Mental Disorder. All analyses were corrected for possible effects of age, gender, and NYHA.

[a]Note: Multiple-choice question.

[b]Note: At least one cell was too small for the appropriate analysis.

## Results

In total, 753 patients were screened for eligibility, and 374 patients were recruited. Overall, ten recruited patients dropped out of the study due to incomplete questionnaires or withdrawal of the informed consent. A total of 364 patients entered the analysis. Separated by treatment setting, we included 107 patients from hospitals, 157 from rehabilitation clinics, and 100 patients in cardiology practices. In the following results, all percentages relate to the maximum number of patients who were eligible for the underlying question. In some cases, this number is equal to the whole sample. Fig 1 depicts a flow chart presenting the recruitment process.

### Sociodemographic and clinical characteristics

Most patients were male ($n$ = 258, 70.9%) with a mean age (both genders) of 65.9 years ($SD$ = 11.4). One-fifth had a left ventricular ejection fraction of <40%. Most patients had a myocardial infarction and were treated with PCI. The HADS-screening was 'positive' in $n$ = 123 patients (33.8%). SCID-I interview revealed that 102 (28.0%) patients had at least one SCID-I diagnosis. For an overview of the characteristics, see Table 1. The most common diagnoses were unipolar depression, anxiety disorder, and substance use/addiction disorder. An overview of all SCID diagnoses is given in the S1 Table.

### Current state of psychosomatic support, diagnostics, and treatment

Approximately 61% of MD patients and 38% of non-MD patients talked to their primary attending physician (e.g., GP or cardiologist) about psychosocial problems. Both MD and non-MD patients reported mostly to only talk 'sometimes' with their physician about MDs. Overall, physicians actively approached 31% of patients to talk about MDs. MD patients were most likely to report a decline in their mental well-being. Forty-six MD patients (45.1%) and six non-MD patients (2.3%) underwent psychological/psychiatric examinations in secondary care. Also, forty-one MD patients (40.2%) knew about their MD diagnosis, and only 28 MD patients (27.5%) received an explanation of what the MD diagnosis means for their further

**Table 3. Estimates of fixed effects, standard errors, p-values, degrees of freedom, and confidence intervals for MD and non-MD patients.**

| Models | $b^a$ | SE | p | df | 95% CI |
|---|---|---|---|---|---|
| **Patients' trajectories** | | | | | |
| Frequency contact with physician in last 4 weeks | -0.03 | 0.191 | .870 | 1 | [-0.406, 0.344] |
| Frequency contact with psychotherapist in last 12 months | -9.53 | 5.393 | .085 | 1 | [-20.410, 1.358] |
| Frequency hospital stays due to heart disease in the last 12 months | 2.21 | 3.100 | .477 | 1 | [-3.898, 8.311] |
| Frequency hospital stays due to other diseases in the last 12 months | -4.57 | 3.631 | .211 | 1 | [-14.649, 0.838] |
| **Frequency of days of sick leave in the last 12 months** | **-19.29** | **7.704** | **.013** | **1** | **[-34.449, -4.138]** |
| Frequency of days of sick leave in the last four weeks | -3.502 | 1.988 | .081 | 1 | [-7.438, 0.434] |
| **Detection rate** | | | | | |
| **The detection rate of MDs in the healthcare system** | **-0.566** | **0.098** | **< .001** | **1** | **[-0.759, -0.373]** |
| **Quality of life** | | | | | |
| Mobility | -0.088 | 0.053 | .100 | 1 | [-0.193, 0.017] |
| Self-care | 0.007 | 0.036 | .840 | 1 | [-0.063, 0.078] |
| **Usual activities** | **-0.139** | **0.058** | **.018** | **1** | **[-0.254, -0.024]** |
| **Pain/discomfort** | **-0.216** | **0.069** | **.002** | **1** | **[-0.352, -0.080]** |
| **Anxiety/depression** | **-0.511** | **0.052** | **< .001** | **1** | **[-0.614, -0.409]** |
| Estimation of current health status | -0.057 | 0.084 | .495 | 1 | [-0.223, 0.108] |
| **Patient satisfaction** | | | | | |
| **Given choices about treatment to think about** | **0.489** | **0.180** | **.007** | **1** | **[0.136, 0.842]** |
| **Satisfied that my care was well organized** | **0.318** | **0.109** | **.004** | **1** | **[0.105, 0.532]** |
| Helped to set specific goals to improve my eating or exercise | 0.300 | 0.157 | .056 | 1 | [-0.008, 0.608] |
| Given a copy of my treatment plan | 0.330 | 0.169 | .052 | 1 | [-0.003, 0.664] |
| Encouraged to go to a specific group or class to help me cope with my chronic condition | 0.207 | 0.186 | .267 | 1 | [-0.159, 0.574] |
| Asked questions, either directly or on a survey, about my health habits | 0.202 | 0.152 | .184 | 1 | [-0.097, 0.501] |
| Helped to make a treatment plan that I could carry out in my daily life | 0.231 | 0.191 | .226 | 1 | [-0.144, 0.606] |
| Helped to plan ahead so I could take care of my condition even in hard times | 0.181 | 0.192 | .345 | 1 | [-0.195, 0.558] |
| Asked how my chronic condition affects my life | 0.057 | 0.183 | .753 | 1 | [-0.302, 0.417] |
| Contacted after a visit to see how things were going | 0.324 | 0.196 | .098 | 1 | [-0.061, 0.709] |
| Told how my visits with other types of doctors, like an eye doctor or other specialist, helped my treatment | 0.208 | 0.187 | .266 | 1 | [-0.159, 0.575] |
| **Total satisfaction** | **0.335** | **0.108** | **.002** | **1** | **[0.123, 0.547]** |

Abbreviations: MD, Mental Disorder. All analyses were corrected for possible effects of age, gender, and NYHA.

[a]Note: Direction of the effect: From MD to non-MD.

treatment. Looking at the treatment section, only 11.3% of all patients (MD and non-MD) received a treatment recommendation, with 6.6% of all patients getting psychotherapy and 4.1% of all patients receiving medication therapy. For detailed information on MD and non-MD patients, please see Table 2.

## Patients' trajectories and detection rate

After correcting for gender, age, and NYHA, significant differences in trajectories of health-care between CHD patients with and without MDs have been found. On average, MD patients reported more days of sick leave over the last twelve months than non-MD patients ($M = 101$ days, SD = 75.9 and $M = 83$ days, SD = 53.8, respectively). Also, it was found that MD patients had more days of sick leave in the last four weeks before the study ($M = 23$, SD = 9.5, and $M = 19$, SD = 9.5, respectively). MD patients also reported making use of psychotherapeutic aid more often than non-MD patients ($M = 10.2$ times, SD = 17.2 and $M = 6.6$ times, SD = 10.1, respectively). However, these differences were only marginally significant.

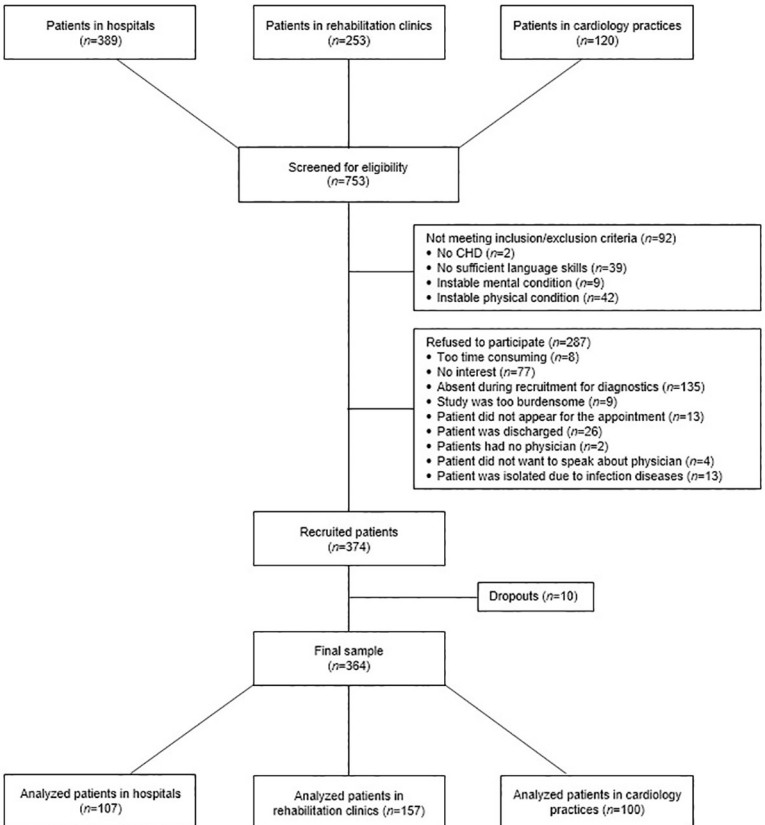

**Fig 1. Flowchart of the recruitment procedure.** Abbreviations: CHD, coronary heart disease.

Overall, MD and non-MD patients were not always correctly diagnosed previously by their physician as such ($\chi^2(1) = 119.412$, $p < .001$). Fifty patients (13.7%, true-positive) were correctly pre-diagnosed with MDs. Further, 255 patients (70.1%) were correctly diagnosed as healthy (true-negative), seven were falsely pre-diagnosed as suffering from MDs (1.9%, false-positive) and 52 (14.3%) were incorrectly pre-diagnosed as healthy (false-negative) with the consequence that these patients suffered from MD symptoms, but did not receive diagnostics nor treatment. Summing up, compared to the total sample, 59 (16.2%) patients were misdiagnosed by their attending, most of which (88.1%) as mentally healthy even though they were not. A total of 102 patients were diagnosed with MDs in this study through our assessment with the HADS and the SCID-I. Fifty of them were correctly pre-diagnosed by the attending physician, yielding in a detection rate of 49.0% on the part of the physicians. Correctly pre-diagnosed MD patients talked more often with their physicians about MDs, were more often actively approached by the physician on MDs, had more psychological examinations, and got more help with the search for psychotherapeutic treatment. Incorrectly pre-diagnosed MD patients were rarely noticed as suffering from MDs, were seldom referred to diagnostics, and rarely got a recommendation for further treatment. For detailed information, see Table 3 and S2 Table.

## Quality of life and patient satisfaction

The analysis of the EQ-5D-3L showed a left-skewed distribution, indicating that most of the participants reported no restrictions in health-related quality of life. Out of the six dimensions,

MD patients differed significantly in three of them. MD patients reported less quality of life when it comes to activities in everyday life than non-MD patients ($M = 1.5$, $SD = 0.6$ vs. $M = 1.3$, $SD = 0.5$, respectively), pain and discomfort ($M = 1.9$, $SD = 0.6$ vs. $M = 1.6$, $SD = 0.6$, respectively) and anxiety and depression ($M = 1.7$, $SD = 0.5$ vs. $M = 1.2$, $SD = 0.4$, respectively). No significant differences have been found on the dimension's mobility, self-care, and general estimation of one's health status.

Overall, patients reported being satisfied with their received medical care across all items ($M = 4.2$, $SD = 0.9$). On average, patients reported high satisfaction with the organization of their care ($M = 4.3$, $SD = 0.9$), the general course of medical treatment ($M = 4.2$, $SD = 0.9$), and receiving a copy of their treatment plan ($M = 4.1$, $SD = 1.4$). Satisfaction was lowest on dimensions of psychological support received by their physician, for example helping the patient develop a plan to be able to cope when the chronic condition worsens ($M = 2.8$, $SD = 1.6$), making a treatment plan to improve daily life ($M = 2.6$, $SD = 1.6$), or encouraging the patient to find self-help groups to cope with their chronic condition ($M = 2.3$, $SD = 1.5$).

MD patients scored (marginally) significantly lower on five patient satisfaction measures as compared to non-MD patients. MD patients stated that, on average, they were given more than one treatment option to choose from less frequently ($M = 2.9$, $SD = 1.5$, and $M = 3.5$, $SD = 1.5$, respectively), they were satisfied less with the organization of their care ($M = 4.1$, $SD = 1.1$, and $M = 4.4$, $SD = 0.8$, respectively), they experienced less support from doctors to set specific goals to improve eating or exercise behavior ($M = 3.6$, $SD = 1.4$, and $M = 3.9$, $SD = 1.2$, respectively), and they reported to have received copies of their treatment plan less frequently ($M = 3.9$, $SD = 1.5$, and $M = 4.2$, $SD = 1.4$, respectively). For more detailed information regarding all items, see Table 3.

## Additional analyses

This study provides some additional analysis regarding the interaction between MDs and healthcare. For detailed information, see S1 Appendix.

## Discussion

The current study aimed to explore (1) the current state of diagnostics and treatment of MD and non-MD in CHD patients, (2) patients' trajectories and detection rate in secondary care, and (3) implications of mental comorbidity and its management on quality of life and patient satisfaction.

### Current state of psychosomatic support, diagnostics, and treatment

Both hypotheses concerning the current state of diagnostics and treatment of MD and non-MD CHD patients were confirmed: It was found that approximately 33.8% of the sample was screened positive on the HADS. SCID-I interview revealed that 28% had at least one SCID-I diagnosis. Only 6.6% of patients underwent psychotherapy and 4.1% medication therapy through psychotherapists/psychiatrists.

First, findings regarding diagnostics are in accordance with the literature, as in general, 30% of patients experience depressive symptoms, 20% fulfill criteria for depressive disorders [5], and most MD patients were diagnosed with depression, anxiety disorder, or a combination of both [4, 5, 7]. Especially depression and anxiety disorders, therefore, have to be taken into account when treating someone for CHD, as they occur regularly, worsen the prognosis [3] and are associated with higher morbidity and mortality [4, 6, 7].

Secondly, the results of the study indicate that there are too few conversations about MD in physician-patient-interactions. This may be one reason why physicians have difficulties to

identify patients with MDs and recommend treatment for MDs. Further, due to known restrictions like lack of time and knowledge [14, 29], it seems following the literature difficult for physicians to address severe issues (e.g., more than MD screening) with psychological content in addition to established somatically-oriented treatment. Also, patients might be insecure about whether they should talk about MDs or are unaware of the option of psychotherapy or drug therapy, which is why the physician should be the one making the first step and ask about possible psychological issues. If physicians suspect that a patient might suffer from MDs, they should provide information about mental healthcare and possibly help patients making contact with psychological caretakers. A beneficial effect would be that patients perceive physicians as empathetic, significantly improving patient satisfaction and compliance in terms of information exchange, satisfaction with treatment, perceived expertise, and interpersonal trust [30]. In the next step, the patient would be referred to a psychotherapist or psychiatrist. This would come with advantages for the patient and the physician alike. The patient would be monitored by an MD expert, resulting in increased diagnostic accuracy and adequate treatment. The physician could concentrate solely on the treatment of CHD and screening for MD-related symptoms.

## Patients' trajectories and detection rate

Regarding patients' trajectories, our expectations were partly confirmed. MD patients utilized parts of the healthcare system more frequently than non-MD patients. MD patients reported being sick more frequently as compared to non-MD patients for the last 12 months. The detection rate of MDs in secondary care (e.g., general practitioners and cardiologists) was about 49%, confirming the hypothesis that a significant amount of MD cases would not be found by physicians in secondary care. The diagnostic accuracy of MDs in CHD patients is insufficient.

It is not surprising that patients with MDs utilize the healthcare system more often than non-MD patients. However, the increased use of the healthcare system seems to occur at the wrong end. Most CHD patients approach their physicians more frequently due to health problems facilitated by their MDs instead of visiting mental healthcare professionals. Physicians are increasingly called upon to provide appropriate MD diagnostics and treatment [31], while physicians' ability to detect, diagnose, and adequately treat patients with MDs is often considered unsatisfied. After three years in the healthcare system, about 14% of patients with depression or anxiety remained unrecognized [32].

The recognition of mental problems in only half of the affected people results in MD patients not getting adequate treatment and possibly worsening their symptoms. The current study adds to the findings that physicians in primary care use significantly fewer procedures to detect and treat MDs than for somatic diseases and that no adequate MD management was provided [15]. Patients with MDs went mostly unrecognized by the physician, indicating that physicians did not screen properly for comorbid MDs in CHD patients. Other studies have shown that 50% of physicians are unaware of MDs like depression to be a risk factor to CHD [14]. Most physicians also did not use any screening tools to detect MDs like depression, nor did they talk to their patients about depressive symptoms. Although the studies mentioned above refer to primary care situations and are therefore not directly comparable with the secondary care examined here, both studies showed that consequences of non-accidental misdiagnoses could be manifold, with limitation of the recovery from and handling of CHD being the most severe consequence. The medical system does not catch a relevant number of patients suffering from MDs. An erroneous detection rate not only hinders patients from addressing CHD-related symptoms properly but also impairs their quality of life in the long turn.

## Quality of life and patient satisfaction

MD patients reported less quality of life on psychological dimensions like discomfort or anxiety, but not on physiological dimensions like mobility or self-care, rendering our expectations regarding quality of life half confirmed. The same holds for the hypothesis regarding MD patients' satisfaction with their treatment. On five of 12 dimensions, MD patients reported a critical shortage concerning aspects of basic psychological care like coping with MD/CHD, and support to gain self-help related skills. Patients expressed a need for unmet psychological support provided by the physician, regardless of whether they suffered from MDs or not.

Patient satisfaction and quality of life seem to be critical components in the treatment of patients with CHD and MDs. Both influence treatment adherence, and as a result, patients who report high satisfaction and quality of life benefit more from care than less satisfied patients. Furthermore, patient satisfaction predicts outcomes right from the initial stages of treatment, e.g., when assessed within the first two days of hospital care [33]. If patients are not satisfied with the care they receive, one has to expect that patients are less likely to adhere to the necessary treatment courses and do not realize recommended lifestyle changes, which can result in a vicious circle in which patients experience more symptoms, get sick more often, have to visit their physician more often, and possibly develop a MD or worsen current MD symptoms [34]. The physician can break this vicious circle by involving the patient in the course of treatment through shared decision-making and patient-centered care (i.a., perceptions of influence on the treatment course, feeling understood by the physician). If this is initiated early enough by the physician within the physician-patient communication, patient satisfaction and thus treatment adherence can be maintained and improved. In the course of shared decision-making, it appears to be beneficial that the physician recommends and discusses in-depth diagnosis and treatment of MDs with the patient after a positive MD screening. The shared decision on MD treatment can also increase adherence, which in turn can improve patient satisfaction and quality of life [35].

## Strengths and limitations

MenDis-CHD gives insights into how CHD patients with MDs are treated in clinical practice and how patients utilized the healthcare system. Another strength of this study was that every person had a thorough psychological screening for depression and anxiety symptoms and, if applicable, an assessment of mental disorders employing SCID-I. Therefore, patients had the opportunity to get a valid diagnosis, immediate support and information about further treatment possibilities.

The current study also had limitations. The sample size on some questions of the questionnaire was too small for statistical analyses due to lack of eligibility, so that analysis remained at a descriptive level. However, only a minority of the self-developed items were affected. Further, we possibly did not detect all MDs because participants were only tested with a SCID-I interview if the HADS-screening was 'positive.' We used a cross-sectional study design in which we examined several cohorts exclusively from the Cologne area, Germany, within a short, limited period. Perhaps the time span or locality may have caused a sample bias because patients were not retested multiple times, and there was no multicenter testing. In addition, subjective items could be affected by recall bias. One example could be the question about the number of days of sick leave during the last year. This could be overcome by adding objective measurements of those items, i.e., the actual, registered number of days of sick leave someone has asked for in the last year.

## Implications for further research

The sample size of the current study was mediocre; therefore, a repetition with a larger sample size would be advantageous to examine the generalizability of the results. The perspective of physicians could be explored regarding their perception of barriers and limitations in care for CHD patients with MDs. Together with the needs and preferences of patients, the results from MenDis-CHD could inform guidelines for the detection and treatment of MDs in CHD patients.

## Conclusion

We found that the state of diagnostics and treatment of MD in patients with CHD was insufficient. Although patients expressed a positive attitude towards addressing MDs within a medical context and were generally satisfied with the medical healthcare, guidance for disease self-management and treatment of MDs was insufficient. Training of physicians concerning screening and treatment of comorbid MDs could have a positive impact on health-related quality of life and possibly could reduce CHD risk factors.

## Supporting information

**S1 Table. Overview of all SCID-I diagnoses.** Abbreviations: SCID, Diagnostic and Statistical Manual of Mental Disorders. [a]Note: patients can have more than one SCID diagnosis.
(DOCX)

**S2 Table. Trajectories of the healthcare system.** Showing the difference in the current state of psychosomatic support between patients who had both a pre-existing MD diagnosis and were positively tested in our study (SCID-I) versus patients who had no pre-existing MD diagnosis but were positively tested (SCID-I). Abbreviations: MD, Mental Disorder; SCID, Diagnostic and Statistical Manual of Mental Disorders. [a]Note: All percentages relate to the maximum number of patients who were eligible for the presented questions. [b]Note: At least one cell was too small for the appropriate analysis.
(DOCX)

**S1 Appendix. Additional analyses: Interaction between MD and healthcare.** Abbreviations: MD, mental disorders.
(DOCX)

## Acknowledgments

Further members of the Cologne Research-Network study group (CoRe-Net; collaboration group): Lena Ansmann, Peter Ihle, Ute Karbach, Ludwig Kuntz, Holger Pfaff, Christian Rietz, Nadine Scholten, Ingrid Schubert, Stephanie Stock, Julia Strupp, and Raymond Voltz.

The lead author for this group is Holger Pfaff, contact e-mail address: holger.pfaff@uk-koeln.de.

We would like to acknowledge the support from the further members CoRe-Net: Jun-Prof. Dr. Lena Ansmann, Department of Organizational Health Services Research, Faculty of Medicine and Health Sciences, Carl von Ossietzky University Oldenburg; Dr. Nadine Scholten and Dr. Ute Karbach, IMVR, Center for Health Services Research Cologne (ZVFK), Faculty of Medicine (FM), FHS, UoC; Prof. Dr. Ludwig Kuntz, Department of Business Administration and Health Care Management, Faculty of Management, Economics and Social Sciences (FMESS), UoC; Prof. Dr. Christian Rietz, Department of Remedial Education, FHS, UoC; Peter Ihle and Dr. Ingrid Schubert, PMV research group, FM, UoC; Prof. Dr. Stephanie Stock,

Institute for Health Economics and Clinical Epidemiology, FM, University Hospital Cologne (UHC); Dr. Dr. Julia Strupp, Department of Palliative Medicine, FM, UHC; Prof. Dr. Raymond Voltz, Department of Palliative Medicine, FM, UHC.

## Author Contributions

**Conceptualization:** Frank Schulz-Nieswandt, Frank Jessen, Christian Albus.

**Data curation:** Samia Peltzer, Hendrik Müller, Ursula Köstler.

**Formal analysis:** Samia Peltzer.

**Funding acquisition:** Frank Schulz-Nieswandt, Frank Jessen, Christian Albus.

**Investigation:** Samia Peltzer, Hendrik Müller, Ursula Köstler.

**Methodology:** Samia Peltzer, Hendrik Müller.

**Project administration:** Samia Peltzer, Frank Jessen, Christian Albus.

**Resources:** Samia Peltzer, Christian Albus.

**Software:** Samia Peltzer.

**Supervision:** Frank Jessen, Christian Albus.

**Validation:** Samia Peltzer.

**Visualization:** Samia Peltzer, Hendrik Müller.

**Writing – original draft:** Samia Peltzer.

**Writing – review & editing:** Samia Peltzer, Hendrik Müller, Ursula Köstler, Frank Schulz-Nieswandt, Frank Jessen, Christian Albus.

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
