## [Decision Letter · Decision Letter 0]

3 Aug 2020

PONE-D-20-20310

Detection and treatment of mental disorders in patients with coronary heart disease (MenDis-CHD): A cross-sectional study

PLOS ONE

Dear Dr. Peltzer,

Thank you for submitting your manuscript to PLOS ONE. After careful consideration, we feel that it has merit but does not fully meet PLOS ONE’s publication criteria as it currently stands. Therefore, we invite you to submit a revised version of the manuscript that addresses the points raised during the review process.

We look forward to receiving your revised manuscript.

Kind regards,

Stephan Doering, M.D.

Academic Editor

PLOS ONE

3. One of the noted authors is a group, CoRe-Net study group. In addition to naming the author group and listing the individual authors and affiliations within this group in the acknowledgments section of your manuscript, please also indicate clearly a lead author for this group along with a contact email address.

Reviewers' comments:

Reviewer's Responses to Questions

**Comments to the Author**

1. Is the manuscript technically sound, and do the data support the conclusions?

Reviewer #1: Yes

Reviewer #2: Partly

2. Has the statistical analysis been performed appropriately and rigorously? 

Reviewer #1: Yes

Reviewer #2: No

3. Have the authors made all data underlying the findings in their manuscript fully available?

Reviewer #1: Yes

Reviewer #2: Yes

4. Is the manuscript presented in an intelligible fashion and written in standard English?

Reviewer #1: Yes

Reviewer #2: No

5. Review Comments to the Author

Reviewer #1: Thank you for the opportunity to review the manuscript titled “Detection and treatment of mental disorders in patients with coronary heart disease (MenDis-CHD): A cross-sectional study”.

The authors aimed to analyze (1) the state of diagnostics and treatment of mental disorders in CHD patients; (2) patient’s trajectories in secondary care, and (3) draw implications regarding quality of life and patient satisfaction.

Regarding their aims, the authors find, that (1) 28% of the sample reported a psychiatric diagnosis (SCID-I), out of which nearly half were correctly pre-diagnosed. Moreover, only 7% underwent psychotherapy and 4% medication therapy for mental disorders; (2) Patients with mental disorders reported longer stays in hospital and more sick days than those without; (3) Patients with mental disorders showed a reduced quality of life and lower patients satisfaction compared to those without.

In sum, this manuscript analyzes an interesting and relevant topic. However, I have some questions and comments regarding methods and reporting, which could help to improve the transparency and readability of the manuscript.

*Overall structure/language

1. In my opinion, because a range of different outcomes was analyzed, and some results were reported on the entire sample, people with mental disorders, or a comparison, the results and discussion could be restructured/revised to enhance the clarity of the manuscript.

2. Moreover, the manuscript could benefit from minor language editing.

*Abstract

The use of bivariate statistics and rather descriptive nature of the study should be stated as early as in the abstract.

*Introduction

1. The authors state that “(...) evidence concerning the quality of psychosomatic care and trajectories within the healthcare system in CHD patients with MD is scarce.” (p. 4). However, there are some longitudinal studies available regarding trajectories of mental health (particularly anxiety and depression), quality of life, mental health treatment, and health care costs in patients with CHD, that are, in my opinion, not considered (e.g. DOI 10.1016/j.jpsychores.2017.10.015; 10.1159/000501502). In light of this, it should be stated more clearly what this study adds to the existing scientific literature.

2. The sentence „MenDis-CHD could successfully archive these aims.“ is, in my opinion, rather a conclusion and should be removed from the aim/introduction section.

*Materials & Methods

1. For reasons of transparency, please elaborate on sampling technique.

2. In general, the description of the questionnaire was rather short and further details on the assessment of outcomes could be added. Please elaborate.

3. Analysis: Because the authors state that only a minority of the questionnaire items had subgroups that were to small for statistical analysis (p. 16), I am particularly interested why the analysis was on a bivariate level and there was no adjustment for confounding variables for the main outcomes, e.g. by means of regression.

*Strength & Limitations:

1. A major strength of this study, that is not mentioned in this section and could be added, is the thorough assessment of mental disorders by means of SCID-I. Having said this, as a positive HADS-screening was the prerequisite for the SCID assessment some MDs may also have been undetected.

2. Moreover, I feel that some statements in this section are a conclusion or result rather than a strength of the study and its design and I would recommend revising.

3. In the limitations section I would strongly recommend to include the cross-sectional nature of the study and the assessment of health care use/sick days for the previous 12 months in self-report, which could be affected by recall bias.

*Tables

1. I was wondering why the results in the tables were presented by recruitment setting, as stratification by setting was not the aim/research question of the study and these results are not discussed in detail.

2. Percentages could be added in Supplementary Tables.

Reviewer #2: Review of manuscript “Detection and treatment of mental disorders in patients with coronary heart disease (MenDis-CHD): A cross-sectional study”

This was a cross-sectional study conducted in three settings (hospital, cardiac rehabilitation (CR) and cardiology practices). From my understanding the aims of the study were to identify the prevalence of mental disorders (MD), by undertaking MD screening; examine the accuracy of MD detection; to investigate the acceptability by patients of being approached by a physician regarding MD; to investigate the prevalence of treatment for MD (presumably amongst those who screen positive); and to comment on the adequacy of the diagnosis and treatment of MD in patients with coronary heart disease (CHD).

There are several issues which I believe need to be addressed before this manuscript is acceptable for publication.

1. There is no clear statement of the aims of the study in the abstract. The aim stated (to explore the state of healthcare recording diagnostics and treatment of MDs in secondary care in CHD patients with MD) is vague and cannot be tested. Three aims are stated in the Introduction, but again these are relatively vague and not clear (eg. aim 2: what ‘patients’ trajectories’ are you referring to? Clear aim needs to be articulated in both the abstract and the Introduction.

2. There are sentences in the Introduction that do not make sense. For example, page 3 line 57-58: ‘…are both among the top ten disorders concerning year’s lives with disability worldwide’ and page 3 line 66 ‘the negative impact of MD on the incidence and prognosis…’ of what? Presumably this is on the CHD incidence and prognosis. The manuscript requires a thorough edit for grammatical correctness.

3. Some of the crucial references are too old and therefore not relevant to today’s setting. For example, page 3 line 76: to support the point that ‘adherence to CHD guidelines in physicians seems to be generally low..’ a reference about acceptance of guidelines by family care physicians in 2002 is cited. It is highly likely that this citation does not reflect guideline acceptance at present.

4. Remove the term ‘suffered from’ in lines 108 and 111 on page 5 and line 185 on page 7. It is more appropriate to say ‘had angiographically documented CHD’ and ‘had severe and unstable physical or mental conditions’ and ‘had had myocardial infarction’. ‘Suffered’ is a subjective word, whereas ‘had’ is objective.

5. In the Assessments section (page 5 lines 118-122), it is unclear whether the 158 item questionnaire was given to all patients or only to those who screened positive for MD. It seems that many of the items would not be relevant unless a person has a diagnosed MD. Indeed, the detail of the 158 item questionnaire is inadequate as the reader is unable to ascertain what is actually asked in this very extensive questionnaire. Has this questionnaire been used before? Who developed it? Has it been validated? Is it publicly available? All this information needs to be provided.

6. In the Results section, on page 8 line 190-191, it is stated that “There were significant differences in trajectories of healthcare between CHD patients with and without MD”. Given that this is a key result of the study (relating to the aims), these results should be presented in a table. At present, the results are presented in sentences referring to days in hospital, days of sick leave etc, with mean number of days reported. However, there is no significance level reported for any of these statements so it is not possible for the reader to determine if these differences were statistically significant. Moreover, in some of these statements the n value is slotted into the sentence, which is not a conventional approach. As stated, this information should be presented comprehensively in a table.

7. Table 1 reports on differences based on patient recruitment setting (hospital, CR, cardiology practice). To my mind, these differences are largely methodological and should be stated in a simple sentence rather than in a table. For example, the sociodemographic differences are largely cofounded: CR participants are younger, more likely living alone and have fewer qualifications, whereas hospital and practice patients are older and more likely retired. By virtue of their younger age, it is not surprising then that the CR cohort have higher rates of anxiety, SCID diagnosis and unipolar depression – these trends are consistent with the literature that reports higher mental health problems in younger patients. Again, I reiterate, these are patterns that are a result of the recruitment strategies used in the study, rather than a key finding relevant to the aims, so they do not deserve to be put in Table 1 but instead should be commented on in describing the heterogeneity of the study sample.

8. Instead of the current Table 1, a table comparing characteristics of MD vs. non-MD patients would be much more revealing to the reader.

9. In the section titled Current state of diagnostics and treatment (page 10), there appear to be some errors. On line 233-234 it is stated that ‘Patients in rehabilitation clinics were most likely to report a decline in their mental wellbeing’, however according to Table 2 this difference was not significant (p=.639). Likewise, the statement on line 238-240 implies that these differences were significant (ie. rehab patients being informed about MD diagnosis, diagnosed more often, received an explanation and got treatment recommendations), whereas according to the data in Table 2 all these differences are not significant. Further, the statement that ‘4.1% had medication therapy’ (line 241) is again not consistent with the data presented in Table 2.

10. Indeed, Table 2 is impossible to interpret. There is no indication throughout the table as to what is the total N for each variable, therefore the numbers (n) and percentages (%) do not correspond. For example, regarding ‘own perception of decline in mental wellbeing’ 20/107 is 18% not 58.8% as reported in the table. Without stating the total N for each variable, it is impossible to verify the data and therefore impossible for the reader to interpret the results.

11. Again, the current Table 2 is not very informative to the reader and does not reflect the aims of the study. The study was not designed to compare the rates of MD and other variables between patients from each of the three study settings: as stated, these findings are merely an incidental bi-product of the methodology employed by the researchers. It would be much more informative to present a table of the differences in all these variables for patients with and without MD.

12. The patient satisfaction data would be better presented in a table, with all relevant questionnaire items shown, so that the reader can quickly scan the results. Currently, there is no actual data shown so it is impossible for the reader to interpret the satisfaction levels for each item, and the relative satisfaction between items.

13. In the section on Interaction between MD and healthcare, why is the sentence reporting differences between rehab and other centres in HADS scores reported here? This belongs back with the earlier data on differences in patient characteristics across the three settings, not here in the section on MD and healthcare.

14. The final results reported on page 13-14 report on differences between MD ad Non-MD patients on talking and not talking to physicians. I cannot understand why this is of any clinical significance? It is completely self-evident that MD patients would talk with physicians about MD more often than would non-MD patients (talking about MD is clearly not relevant to someone who does not have MD). Can the authors explain the clinical relevance of this information, or exclude it?

15. In the Discussion, line 316 on page 14, it is stated that women were twice as likely to develop MDs. However, this has not been reported or mentioned in the Results (or I was unable to find it). Plus, why is this result refenced to citation number 26 if it is from the findings of the present study?

16. Overall, the Discussion needs to be presented in light of previous literature and studies regarding the detection and management of mental disorders in cardiac patients, and in the population more broadly. In its current form, it is not adequately nuanced, it does not adequately relate the study findings to the literature, it is not synthesised in a way which adds further illumination of the findings for the reader, and is largely a summary of some of the findings presented in the Results.

17. Some of the references are incomplete. For example, references 8, 10, 20,

6. PLOS authors have the option to publish the peer review history of their article (what does this mean?). If published, this will include your full peer review and any attached files.

Reviewer #1: No

Reviewer #2: No

---

## [Author Response · Author response to Decision Letter 0]

6 Oct 2020

We added further information regarding data availability in the revised cover letter. Every other changes were marked in the new uploaded manuscript and in the point-by-point correction for the PLoS One editor and the two reviewers

---

## [Decision Letter · Decision Letter 1]

30 Oct 2020

PONE-D-20-20310R1

Detection and treatment of mental disorders in patients with coronary heart disease (MenDis-CHD): A cross-sectional study

PLOS ONE

Dear Dr. Peltzer,

Thank you for submitting your manuscript to PLOS ONE. After careful consideration, we feel that it has merit but does not fully meet PLOS ONE’s publication criteria as it currently stands. Therefore, we invite you to submit a revised version of the manuscript that addresses the points raised during the review process. Please check reviewer 1´s comments in particular. I agree, that your manuscript still needs some clarification and improvement of readability.

We look forward to receiving your revised manuscript.

Kind regards,

Stephan Doering, M.D.

Academic Editor

PLOS ONE

Reviewers' comments:

Reviewer's Responses to Questions

**Comments to the Author**

1. If the authors have adequately addressed your comments raised in a previous round of review and you feel that this manuscript is now acceptable for publication, you may indicate that here to bypass the “Comments to the Author” section, enter your conflict of interest statement in the “Confidential to Editor” section, and submit your "Accept" recommendation.

Reviewer #1: (No Response)

Reviewer #2: All comments have been addressed

2. Is the manuscript technically sound, and do the data support the conclusions?

Reviewer #1: Partly

Reviewer #2: Yes

3. Has the statistical analysis been performed appropriately and rigorously? 

Reviewer #1: Yes

Reviewer #2: (No Response)

4. Have the authors made all data underlying the findings in their manuscript fully available?

Reviewer #1: Yes

Reviewer #2: Yes

5. Is the manuscript presented in an intelligible fashion and written in standard English?

Reviewer #1: No

Reviewer #2: Yes

6. Review Comments to the Author

Reviewer #1: Thank you for the opportunity to review the revised manuscript titled “Detection and treatment of mental disorders in patients with coronary heart disease (MenDis-CHD): A cross-sectional study”.

I would like to thank the authors for all the important revisions made to the manuscript. While this has enhanced the readability of your work, there are still some uncertainties, which, I assume, result from unclear wording and the many outcomes/items analyzed in this manuscript. I have provided some suggestions to enhance transparency to the reader. However, further revision and editing may be needed as well.

1. General notes on wording

1.1. Please make sure to use abbreviations/wording uniformly to enhance readability: “EQ5D” vs. “EQ-5D”;“CHD patients” vs. “CHD-patients”; for mental disorders, sometimes “MD” is used for singular and plural forms, while at other times, “MDs” is used to indicate the plural form; “sick leave” vs. “days of illness”, etc.

1.2. “Trajectory”: In my opinion “trajectory” refers to the observation of individuals at several points in time which would permit the analysis of intra-individual change or trajectories over time using longitudinal datasets. Because this was a cross-sectional study with a retrospective assessment of preceding health care use, this should be made clearer, e.g. “previous health care use and days of sick leave within the last X months”.

1.3. “Diagnosis”: Particularly in the section “patients’ trajectories and detection rates” it is oftentimes not quite clear what the term “diagnosis” refers to - does it mean MD/SCID result in your study or a previously documented diagnosis, i.e. detection as a case by a previous physician? Please revise for clarity.

2. Introduction

I originally suggested to state the relevance of the study in more detail as well as in light of some more recent longitudinal studies that were, in my opinion, not considered when considering health care use/ costs/quality of life associated with mental health problems and depression management in patients with CHD in the introduction (“originally p. 4, “(...) evidence concerning the quality of psychosomatic care and trajectories within the healthcare system in CHD patients with MD is scarce.”). While the authors stated to have included the references and discussed the difference to their study, I cannot find them in the revised manuscript.

3. Methods

Thank you for providing further information on the assessments, Unfortunately, even with the added details, it is still not clear to me which assessment and which statistical method was used for which reported result exactly. In line with this, by which assessment was the detection rate of mental disorders in healthcare assessed?

Further restructuring of the “Assessments”-Section by section headings could be helpful. Moreover, the exact method of analysis for each research question could be added to the Tables/briefly mentioned in the text; or each research question and method of analysis should be described in more detail in the “data analysis” section for reasons of transparency.

4. Results

4.1. Please avoid the use of “marginally significant”. Because an alpha of 0.05 was set for all analyses as the threshold for significant effects, results are either significant at this threshold or they are not.

In line with this, why are p-values over 0.05 (e.g., contact with psychotherapists: p = 0.085) printed in bold in Table 3?

4.2. Additional analyses on interaction: The consideration of the different recruitment settings was not described as an aim of the study, there seems to be no a priori hypothesis about this additional analysis, and the results are not discussed in the discussion section. In line with this and because the manuscript already includes many analyses and outcomes, I would suggest to fully exclude the additional analysis or put it completely into the appendix as an exploratory, additional analysis, with a brief reference to it in the main manuscript.

5. Discussion

5.1. As far as I can tell, the study could not analyze whether treatment received for MDs was adequate according to treatment guidelines, such as the German S3-Leitlinien (i.e., details on type and dose of medication/detailed information on psychotherapy). However, treatment adequacy was discussed (l. 396), while the study analyzes to whether or not the physician talked about mental health with their patients? Please revise.

5.2. In the strengths & limitations section it is described that participants who fulfilled the criteria for a SCID-diagnosis were offered immediate psychological support and further information on psychotherapeutic treatment. While this thoughtful offer seems to be part of the study design (and should therefore be mentioned in the methods section) it does not seem to immediately influence the results of this manuscript or their interpretation. Therefore, it should not be described as detailed in the strengths section.

Reviewer #2: The authors have adequately addressed the reviewers' comments and the manuscript is now much improved.

7. PLOS authors have the option to publish the peer review history of their article (what does this mean?). If published, this will include your full peer review and any attached files.

Reviewer #1: No

Reviewer #2: No

---

## [Author Response · Author response to Decision Letter 1]

16 Nov 2020

Thank you very much for your helpful recommendations. We adapted the manuscript following your recommendations and hope to improve the quality of it.

---

## [Editor Report · Decision Letter 2]

26 Nov 2020

Detection and treatment of mental disorders in patients with coronary heart disease (MenDis-CHD): A cross-sectional study

PONE-D-20-20310R2

Dear Dr. Peltzer,

We’re pleased to inform you that your manuscript has been judged scientifically suitable for publication and will be formally accepted for publication once it meets all outstanding technical requirements.

Kind regards,

Stephan Doering, M.D.

Academic Editor

PLOS ONE

---

## [Editor Report · Acceptance letter]

4 Dec 2020

PONE-D-20-20310R2 

Detection and treatment of mental disorders in patients with coronary heart disease (MenDis-CHD): A cross-sectional study 

Dear Dr. Peltzer:

I'm pleased to inform you that your manuscript has been deemed suitable for publication in PLOS ONE. Congratulations! Your manuscript is now with our production department. 

Kind regards, 

on behalf of

Professor Stephan Doering 

Academic Editor

PLOS ONE